# Immune Modulation Through KIR–HLA Interactions Influences Cetuximab Efficacy in Colorectal Cancer

**DOI:** 10.3390/ijms26168062

**Published:** 2025-08-20

**Authors:** María Gómez-Aguilera, Bárbara Manzanares-Martín, Arancha Cebrián-Aranda, Antonio Rodríguez-Ariza, Rafael González-Fernández, Laura del Puerto-Nevado, Jesús García-Foncillas, Enrique Aranda

**Affiliations:** 1Medical Oncology Department, Reina Sofia University Hospital, University of Cordoba, 14071 Cordoba, Spain; maria.gomez.aguilera.sspa@juntadeandalucia.es (M.G.-A.); earandaa@seom.org (E.A.); 2Department of Immunology and Allergy, Reina Sofía University Hospital, 14004 Cordoba, Spain; barbara.manzanares.sspa@juntadeandalucia.es (B.M.-M.); rafael.gonzalez.sspa@juntadeandalucia.es (R.G.-F.); 3Maimónides Biomedical Research Institute of Cordoba (IMIBIC), Reina Sofía University Hospital, University of Cordoba, 14071 Cordoba, Spain; 4Translational Oncology Division, Oncohealth Institute, Hospital Universitario “Fundación Jimenez Diaz”, Autonomous University of Madrid, 28040 Madrid, Spain; arancha.cebrian@quironsalud.es (A.C.-A.); lpuerto@quironsalud.es (L.d.P.-N.); jgfoncillas@quironsalud.es (J.G.-F.); 5Cancer Network Biomedical Research Center (CIBERONC), Av. Monforte de Lemos, 28029 Madrid, Spain

**Keywords:** KIR, anti-EGFR, cetuximab, metastatic colorectal cancer, natural killer cells, HLA ligands

## Abstract

Colorectal cancer (CRC) remains a major cause of cancer-related mortality. Cetuximab improves survival by combining EGFR inhibition with immune activation. This study evaluated the influence of killer cell immunoglobulin-like receptor (KIR)-mediated immune responses on cetuximab efficacy in 124 metastatic CRC patients: 55 with wild-type (WT) KRAS and 69 with KRAS mutations. Peripheral blood was genotyped for 19 KIR genes and relevant HLA alleles, focusing on key KIR–HLA interactions (2DL1–C2, 3DL1–Bw4, 3DS1–Bw4). KRAS-WT patients showed better outcomes, receiving more treatment cycles (median: 17 vs. 4) and showing slower disease progression (60% vs. 92.8% at 12 months). WT patients had higher frequencies of inhibitory KIRs and the Bw4 allele, with KIR3DS1–Bw4 heterozygosity linked to longer survival (*p* = 0.013). In KRAS-mutant patients, heterozygous KIR genotypes (AB) and mixed A/B semi-haplotypes were associated with improved survival (*p* = 0.002). Multivariate analysis confirmed KIR3DS1–Bw4 as a favorable factor in WT patients and AB genotypes as beneficial in KRAS-mutants. In conclusion, KIR–HLA interactions significantly impact cetuximab efficacy in metastatic CRC, with distinct immunogenetic profiles in WT and KRAS-mutant patients. These results highlight the potential of KIR–HLA profiling to guide personalized treatment strategies.

## 1. Introduction

Colorectal cancer (CRC) is the second leading cause of cancer-related deaths worldwide [1,2]. However, survival rates can be improved by optimizing treatment approaches, particularly through a deeper understanding and more effective use of monoclonal antibodies (MoAbs) like cetuximab [3]. Cetuximab is an IgG1 isotype MoAb, and this is significant because IgG1 antibodies have a unique ability to engage both tumor cells and the immune system. Thus, cetuximab, exerts its effects through two primary mechanisms. First, its antigen-binding fragment (Fab) specifically binds to the epidermal growth factor receptor (EGFR), inhibiting its activity and blocking the subsequent intracellular signaling pathways within tumor cells. This disruption prevents cell proliferation and metastasis [4,5]. However, this mechanism is only effective in patients with KRAS wild-type colorectal cancer, as those with KRAS mutations have constitutively active intracellular signaling, rendering cetuximab ineffective [6,7]. Second, cetuximab’s crystallizable fragment (Fc) binds to the CD16 receptor on immune cells, triggering an immunomodulatory response that facilitates tumor cell lysis [8,9]. This action is solely dependent on the type of immunoglobulin (IgG1) and not on KRAS status, thus functioning similarly across all IgG1 MoAbs like cetuximab.

The Fc region of IgG1 antibodies binds to Fcγ receptors on various immune cells, including natural killer (NK) cells, macrophages, and neutrophils. Specifically, it binds to CD16 (FcγRIII) on NK cells, which is critical for initiating a process known as antibody-dependent cellular cytotoxicity (ADCC). The function of NK cells in ADCC is tightly regulated by a range of molecules, notably the killer cell immunoglobulin-like receptors (KIRs) [10]. When NK cells encounter a target cell, they integrate signals from both activating and inhibitory receptors, responding quickly if the activating signals dominate [11]. By carefully integrating these signals, NK cells contribute to the immune system’s ability to target abnormal cells without causing unnecessary damage to normal tissues.

The cytotoxicity mediated by KIR is dependent on their interaction with HLA ligands present on tumor cells [12]. Both KIR and HLA exhibit a high degree of genetic polymorphism and are inherited independently, which increases the range of potential immune responses. This diversity allows the immune response to adapt quickly to rapidly changing environments, such as those encountered in cancer [9,13]. Research has shown that NK cell effector mechanisms are significantly enhanced when activation occurs through the CD16 receptor, which interacts with the Fc region of IgG1 isotype antibodies, alongside the interaction between KIRs and their HLA ligands on tumor cells [14,15,16]. This dual activation can lead to a more potent antitumor effect of cetuximab, allowing for a clearer identification of immune effector mechanisms that may otherwise remain obscured in the absence of this treatment. Previous research conducted by our group on various tumor types has indicated that specific genetic profiles, such as KIR alone or KIR–HLA combinations, can serve as predictors for responses to cetuximab and trastuzumab [17,18]. However, the precise role of KIR and their HLA ligands in the effectiveness of these drugs remains unclear. There is currently insufficient data to determine whether the KIR effect is solely attributable to cetuximab’s immunomodulatory function (mediated by the Fc region), which could apply to other IgG1 MoAbs, or if it results from the synergistic action of both the Fab and Fc mechanisms. Since Fc activation is independent of KRAS status, it is postulated that in patients with KRAS mutations who respond to cetuximab, this mechanism is the primary contributor. Consequently, these patients could provide a unique “in vivo” model to clarify the role of KIR-mediated response in conjunction with IgG1 isotype MoAbs [6,7,19,20,21,22].

Based on this, the objective of this study is to compare the genetic KIR profiles predictive of cetuximab response between CRC patients with KRAS mutations and those with wild-type (WT) KRAS. This comparison aims to determine whether the KIR effect in cetuximab-treated WT patients is solely dependent on the Fc region, to assess how cetuximab efficacy relates to the KIR-mediated immune response.

## 2. Results

### 2.1. Baseline Patient Characteristics and Clinical Outcome

The baseline clinical characteristics of KRAS-WT and KRAS-mutant patients are summarized in Table 1. The population was demographically homogeneous in terms of geographic origin, ethnicity, and other sociodemographic variables. Median age at enrollment was 64 years in both groups. Male predominance was higher in the KRAS-WT group (70.9%) than in the KRAS-mutant group (50.7%). Most patients in both groups had a baseline ECOG performance status of 0 or 1, indicating a good functional status.

Notably, KRAS-WT patients received significantly more treatment cycles than KRAS-mutant patients (median: 17 vs. 4), reflecting a better response to cetuximab. Disease progression within 12 months occurred in 60% of KRAS-WT patients (33/55) versus 92.8% (68/69) of KRAS-mutant patients highlighting the strong influence of KRAS status on treatment outcomes.

### 2.2. Frequencies of KIR Genes and Their Ligands

We first analyzed KIR and HLA gene frequencies in both groups to identify genetic profiles potentially influencing cetuximab response (Table 2). Allele distributions were consistent with data from the Allele Frequency Net Database (http://www.allelefrequencies.net, accessed on 23 April 2025) [23] KIR2DL4, KIR3DL2, KIR3DL3, and KIR3DP1 were present in all patients, and excluded from further analysis due to their ubiquity.

Although differences were not statistically significant, variations in gene distribution were observed. Inhibitory KIRs (2DL1, 2DL2, 2DL3, 2DL5, and 3DL1) were more frequent in WT patients. Among activating KIRs, KIR2DS2, KIR2DS5, and KIR3DS1 were more common in WT patients, while KIR2DS1, KIR2DS3, and KIR2DS4 were more frequent in KRAS-mutant patients. The pseudogene KIR2DP1 was also more prevalent in WT. For HLA ligands, Bw4 and C1 alleles were more frequent in WT, whereas C2 was more common in the mutant group.

### 2.3. Frequency of KIR–Ligand Combinations

KIR–HLA combinations influence NK cell activation; therefore, we analyzed their distribution (Table 3). While differences were not statistically significant, WT patients showed higher frequencies of KIRs with Bw4 (3DL1, 3DS1) and C1 ligands (2DL2, 2DL3), suggesting stronger NK activity. In contrast, KIRs with C2 ligands (2DL1, 2DS1) were more common in KRAS-mutant patients, potentially indicating altered NK responses.

### 2.4. Frequency of KIR Genotypes and Haplotypes

Table 4 shows KIR genotypes and haplotypes. The heterozygous AB genotype (one inhibitory, one activating), was more frequent in KRAS-WT patients, while homozygous AA or BB genotypes were more common in KRAS-mutant patients. Regarding haplotype structure, combinations with both activating or both inhibitory components were more frequent in the mutant group, whereas mixed haplotypes (one activating, one inhibitory) were more common in WT patients.

### 2.5. Survival Analysis According to KIR Genes or Their Ligands

After confirming similar KIR and HLA ligand frequencies in both groups, we analyzed their impact on progression-free survival at 12 months (PSF12) following cetuximab treatment (Table 5). In KRAS-WT patients, most KIRs and ligands were linked to longer PFS12, with significant associations for KIR2DL1 and KIR2DP1 (*p*-value = 0.031), and a trend for Bw4 (*p*-value = 0.064). Conversely, KIR2DS4, KIR3DL1, and C1 were associated with poorer outcomes. In KRAS-mutant patients, few differences were observed, except for KIR2DL5, which was associated with longer PFS12 (*p*-value = 0.032) (Table 5, Figure 1).

### 2.6. Survival Analysis According to KIR–Ligand Combinations

Survival analysis of KIR–ligand combinations showed no significant differences in KRAS-mutant patients. However, in KRAS-WT patients, the KIR3DS1–Bw4 combination was significantly associated with longer PFS12 (*p*-value = 0.031) (Figure 2A). Among those with KIR3DS1, Bw4Bw6 heterozygotes had better outcomes than homozygotes (*p*-value = 0.013) (Figure 2B).

### 2.7. Survival Analysis According to KIR Haplotypes

PFS12 analysis by KIR haplotypes (AA, AB, BB) showed no survival differences in KRAS-WT patients. However, in KRAS-mutant patients, the AB genotype (one activating, one inhibitory haplotype) was linked to significantly longer PFS12 (*p*-value= 0.002) (Figure 3). AB patients had better outcomes than both AA (*p*-value = 0.001) and BB (*p*-value = 0.03) genotypes, with AA showing the shortest survival and BB intermediate.

### 2.8. Survival Analysis According to Centromeric–Telomeric KIR Haplotypes

Analysis of centromeric and telomeric KIR haplotypes (Table 6) showed no survival differences in KRAS-WT patients. Conversely, KRAS-mutant patients with mixed haplotypes (one A and one B) had significantly longer PFS12 (*p*-value = 0.002), regardless of location (CEN versus TEL). The poorest outcomes were seen in those with two inhibitory haplotypes (CENA/TELA) (*p*-value = 0.011, Table 6).

### 2.9. Univariate and Multivariate Analyses

Univariate and multivariate analyses identified factors independently associated with progression risk. In KRAS-WT patients (Table 7), both the number of treatment cycles and the KIR3DS1–Bw4 combination were significant. Bw4Bw6 heterozygotes had the best outcomes, with a 51% reduced risk of progression (HR = 0.49). In contrast, HLA-B homozygosity with KIR3DS1 increased risk of progression (HR = 2.22). No other KIR or HLA markers, including haplotypes or genotypes, showed significant associations.

In the multivariate analysis of KRAS-mutant patients (Table 8), significant factors were the number of metastatic sites, genotype (AA/BB vs. AB), and B semi-haplotypes (CENA/TELB or CENB/TELA versus CENA/TELA). Homozygous AA or BB genotypes were linked to a two-fold higher risk of progression compared to AB. No significant effects were observed for individual KIRs or their ligand interactions.

## 3. Discussion

This study provides the first comparative analysis of KIR profiles and their HLA ligands in two distinct groups of mCRC patients treated with cetuximab: those with KRAS mutations and those with KRAS wild-type tumors. In a previous study, we provided evidence of the immunomodulatory effects of cetuximab in mCRC patients with a KRAS mutation, although that analysis focused solely on KIR genotypes [24]. The current study expands on this by integrating both KIR and HLA ligand data, and by comparing genetic patterns grouped as semi-haplotypes, haplotypes, and genotypes in KRAS-WT versus KRAS-mutant patients. Rather than aiming to define predictive profiles, our objective was to compare these two populations and determine if they share a common profile, exploring whether KIR gene impact on the response to cetuximab extends beyond its Fc-mediated immunomodulatory effect to include Fab-related mechanisms as well.

Among the immunomodulatory mechanisms involving the Fc region, its ability to bind to immune cells with specific receptors is particularly important. This includes binding to NK cells, which can enhance the anti-tumor response through their activating and inhibitory KIR receptors [25]. Currently, it remains unclear whether blockade of EGFR with cetuximab could influence the selection of different response profiles by reducing tumor mass, releasing neoantigens, or altering HLA expression. In this context, it has been suggested that combining NK cells with MoAbs could accelerate tumor cell death due to the additive effects of both target-specific blockade with the MoAb and the potent cytotoxic activity of NK cells [18].

Our initial finding was that KIR and HLA gene frequencies were comparable between KRAS-WT and KRAS-mutant patients, minimizing the risk of confounding in survival analyses. These findings are particularly noteworthy given that no published studies to date have directly compared KIR–HLA profiles between KRAS-WT and KRAS-mutant patients. Moreover, there is a notable lack of research on how KIR and HLA influence survival in both KRAS wild-type and KRAS-mutant patients following cetuximab treatment. Most existing studies on KIR–HLA in cancer have focused primarily on an individual’s susceptibility to developing a particular tumor, rather than on survival or response to specific treatments [26]. This underscores the novelty and clarity of our approach.

Al Omar et al. [27] reported that five activating KIR genes (2DS1, 2DS2, 2DS3, 2DS5, and 3DS1) were more frequent in patients with colorectal cancer (CRC), with the highest risk associated with the 3DS1 gene, followed by the 2DS1 gene, while 2DS2 showed an inverse relationship with CRC risk [27]. However, this study was included in a meta-analysis that found a correlation only with 2DS5 and no protective KIR gene was confirmed [28]. In a recent study, Diaz-Peña concluded that the prevalence of 2DS3 was significantly higher in CRC patients than in healthy controls [29]. On the other hand, Portela et al. [30] reported that individuals carrying the Bx haplotype might have partial protection against CRC. However, their hypothesis suggested that this relationship was not with a specific KIR or HLA ligand but rather with the presence of various activating KIRs [30]. In our study, the frequencies of 2DS5, 2DS3, and Bx haplotypes did not differ significantly between KRAS-WT and KRAS-mutant groups or healthy controls, suggesting that cetuximab response is more likely related to its mechanism of action, rather than on the presence of specific disease-related genetic profiles.

The analysis of KIR genes in patients with KRAS mutations revealed a significant association between KIR2DL5 and increased PFS12, although this association was not significant in the multivariate analysis. Additionally, neither HLA ligands nor KIR–HLA combinations had an impact on PFS12 of KRAS-mutant patients. In contrast, we observed that in KRAS-WT patients, a significant association with improved PFS12 was present in individuals carrying the KIR3DS1–Bw4 combination, indicating that this KIR–HLA pairing may serve as a predictor of response to cetuximab. A supporting factor for our findings is that KIR3DS1 is associated with both homozygosity and heterozygosity for Bw4, and under normal conditions, this activating receptor is inhibited in NK cells to prevent autoimmune responses [31]. Therefore, Bw4Bw6 heterozygotes are more likely to efficiently activate NK cells via KIR3DS1, which aligns with our observation that patients with the KIR3DS1–Bw4Bw6 combination experienced greater survival following cetuximab treatment than those with KIR3DS1–Bw4Bw4 or KIR3DS1–Bw6Bw6. Although it has yet to be definitively proven that Bw4 is the ligand for KIR3DS1, there is substantial evidence supporting this hypothesis, suggesting functional advantages to their concurrent presence [32].

In a previous study, we suggested KIR2DS4 as a potential prognostic marker in KRAS-mutant patients, though that analysis involved a different population and focused on overall survival. Hence, its predictive value for PFS12 in the context of cetuximab remains uncertain. Another study reported that different combinations of KIR and HLA might help predict overall survival in metastatic colorectal cancer (mCRC) patients who received chemotherapy. Specifically, multivariate analysis identified treatment type and the combination of KIR3DL1–HLA–Bw4-I80 as independent parameters associated with overall survival, although no significant KIR–HLA associations were found for time to progression [33]. Currently, no KIR or KIR–HLA combination has been firmly established as a predictor of cetuximab response, although several studies highlight the role of NK cell-mediated cytotoxicity in MoAb therapy [34,35,36,37,38].

Terszowski et al. [39] investigated KIR–HLA interactions in lymphoma patients treated with MoAbs, specifically rituximab and obinutuzumab. Although both antibodies target CD20, obinutuzumab has been modified to enhance its ADCC activity. The study found that KIR–HLA interactions exhibit different behaviors for these two antibodies, which may lead to varying response profiles following the initial activation of NK cells through Fc binding [39,40]. Other studies have demonstrated that IgG1 MoAbs can activate NK-mediated ADCC. For example, rituximab enhances NK cell responses against B-cell leukemias, ADCC can be triggered in neuroblastomas through anti-GD2 antibodies, and daratumumab (anti-CD38) is utilized in multiple myeloma [41,42,43,44,45]. However, these studies do not assess the potential role of specific target blockade in contributing to this enhanced NK activity.

We also analyzed the impact of KIR genotypes, haplotypes, and semi-haplotypes on PFS12. In a previous study conducted by our group with KRAS-mutant mCRC patients, multivariate analysis confirmed that the classification of patient genotype served as an independent marker of PFS12, with the centromeric and telomeric distribution of KIRs also acting as independent predictors of both PFS12 and overall survival [24]. Here, we observed that such associations were stronger in KRAS-mutant patients, supporting a greater Fc-mediated effect of cetuximab in this group. In contrast, no clear genotype or haplotype effect was found in KRAS-WT patients, where cetuximab is more likely to exert combined Fab and Fc effects. Interestingly, while TELB haplotype has been linked to better response to trastuzumab in HER2+ breast cancer [46], we found no such association in KRAS-WT CRC patients treated with cetuximab.

Our findings suggest that in KRAS-WT patients, the therapeutic benefit of cetuximab may result from both EGFR blockade and immune activation via Fc binding. Blocking EGFR might reshape the tumor microenvironment, enhancing NK cell function through cytokine release (e.g., IL-12, IL-15) and interferon-gamma [47]. These mechanisms can potentially result in the expression of novel tumor antigens, boost HLA expression, and increase the susceptibility of tumor cells to destruction. At the same time, EGFR inhibition may help suppress cell proliferation and metastasis, creating a context in which KIR-mediated responses become more relevant.

In summary, our results suggest that no common KIR–HLA response pattern to cetuximab exists between KRAS-WT and KRAS-mutant groups. In KRAS-WT patients, the KIR effect likely depends on both Fc-mediated activation and EGFR inhibition. This suggests that blocking EGFR may alter the immune response, potentially affecting the predictive value of the KIR–HLA profile in mCRC patients with KRAS-WT. This implies that the interaction between immune regulation and EGFR signaling may influence how the KIR–HLA profile predicts treatment outcomes in KRAS-WT metastatic colorectal cancer patients. These findings warrant further research to explore the underlying mechanisms of this relationship and to determine how they can be leveraged to improve treatment strategies and patient outcomes.

## 4. Materials and Methods

### 4.1. Patients and Inclusion Criteria

A total of 124 patients diagnosed with metastatic CRC (mCRC) were selected for this study, comprising 69 with KRAS mutations and 55 with wild-type KRAS. Patients with KRAS wild-type were recruited from the Medical Oncology Unit at Reina Sofía University Hospital, Cordoba, Spain, between June 2013 and October 2015. The selection criteria included: (a) age over 18 years, (b) signed informed consent, (c) diagnosis of metastatic colorectal cancer, (d) absence of KRAS mutation, and (e) receipt of anti-EGFR therapy with cetuximab. The 69 patients with KRAS mutations were derived from a Phase II clinical trial conducted at Jiménez Díaz Foundation Hospital, Madrid, Spain. The design and methods of this multicenter clinical trial have been published previously [48]. For our study, we included 69 patients from this trial, excluding those for whom sufficient data or DNA for analysis were not available. As this was a retrospective case-control study based on previously defined patient cohorts, no prior sample size calculation was performed. Including the full dataset aimed to ensure a comprehensive analysis and strengthen the validity of the results.

### 4.2. Ethical Considerations

This study adheres to the highest standards of good clinical practice and aligns with the principles outlined in the Declaration of Helsinki. Approval for the study was obtained from the Córdoba Ethics Committee in May 2013, as well as from the Institutional Ethics Committee of Fundación Jiménez Díaz University Hospital, under authorization number EC 02-12 IIS-FJD. To protect patient confidentiality, all personal information was anonymized, ensuring that individual identities remain undisclosed. Data collection and storage were conducted in strict accordance with the current Spanish Data Protection Act (Organic Law 3/2018 regarding the Protection of Personal Data and guarantees of digital rights), safeguarding the rights and privacy of all participants involved in the study.

### 4.3. Sample Processing and DNA Isolation for Genotyping

Peripheral blood samples were collected from each patient using Vacutainer tubes containing Acid Citrate Dextrose (ACD) to prevent coagulation. Following collection, the samples were stored at 4 °C until processing, which was conducted within 24 h to ensure optimal DNA quality. All samples were labeled with unique identifiers to maintain patient confidentiality, and appropriate measures were taken to ensure compliance with ethical standards regarding sample handling. Genomic DNA was extracted from blood samples using the MagCore Super Automated Nucleic Acid Extractor (RBC Bioscience, New Taipei City, Taiwan). DNA samples were then stored at −20 °C for long-term preservation, ensuring that high-quality material was available for subsequent genotyping analyses, including KIR and HLA typing. KIR and HLA genotyping were performed in our laboratory, accredited by the European Federation for Immunogenetics (EFI), following internationally recognized standards for Histocompatibility and Immunogenetics testing. Each genotyping run included internal positive and negative controls to monitor assay performance and specificity. Accuracy was further verified through participation in the GECLID external quality assessment program for both HLA typing and KIR genotyping, sponsored by the Spanish Society of Immunology and the Iberian Society of Cytometry. In this program, the laboratory has consistently achieved a genotyping error rate of 0% over the past several years, confirming the robustness and reliability of the typing procedures.

### 4.4. KIR Genotyping

KIR genes were analyzed using the “KIR Ready gene” kit from Inno-train Diagnostik GmbH, Kronberg, Taunus, Germany (Catalog number: 002 060 040). This technique allows for the identification of the following KIR genes: 2DL1, 2DL2, 2DL3, 2DL4, 2DL5, 2DS1, 2DS2, 2DS3, 2DS4, 2DS5, 3DL1, 3DL2, 3DL3, 2DP1, 3DP1, and 3DS1. Based on the KIR gene content, haplotypes are categorized as either A (inhibitory) or B (activator). The A haplotype is characterized by the presence of the following genes: 3DL3, 2DL3, 2DP1, 2DL1, 3DP1, 2DL4, 3DL1, 2DS4, and 3DL2. In contrast, the B haplotype comprises any combination of KIR genes that excludes those of the A haplotype [49,50] Genotypes were classified as Genotype AA if homozygous for the A haplotype, and Genotype Bx, where x can be either A or B. Additionally, KIR haplotypes consist of two regions: centromeric (CEN) and telomeric (TEL), each with distinct gene content. The CENA motif corresponds to the CEN region of the A haplotype, while TELA corresponds to its TEL region. In contrast, CENB and TELB represent the CEN and TEL motifs of the B haplotype, respectively. Specifically, CENA is defined by the presence of 2DL3; CENB includes 2DS2, 2DL2, 2DL5B, and 2DS3; TELA is characterized by 3DL1 and 2DS4; and TELB is defined by 3DS1, 2DL5A, 2DS5, and 2DS1 (49).

### 4.5. HLA Genotyping

To differentiate the KIR ligands present in each patient, we identified the HLA-B and C loci using the following typing kits from Lifecodes (Immucor Medizinische Diagnostik GmbH, Dreieich, Germany): LIFECODES HLA-A SSO Typing Kit (Catalog number: 628911), LIFECODES HLA-B SSO Typing Kit (Catalog number: 628915), and LIFECODES HLA-C eRES SSO Typing Kit (Catalog number: 628921). In cases of ambiguity regarding allele assignment, next-generation sequencing (NGS) typing was performed as a supplementary method. For this, the NGSgo-MX6 multiplex amplification method (GenDx, Utrecht, The Netherlands) was utilized, followed by sequencing on the Illumina Myseq platform. Genotyping was conducted at a resolution sufficient to determine Bw4 specificities and categorize HLA-C allotypes into C1 and C2 groups.

### 4.6. KIR–HLA Interactions

In our analysis of KIR–HLA interactions, we focused on several well-documented interactions in the literature that hold functional significance in immune responses [9,51,52]. Specifically, we examined the following pairings: 2DL1–C2, 2DL2/3–C1, 3DL1–Bw4, 2DS1–C2, and 3DS1–Bw4.

### 4.7. Statistical Analysis

A descriptive analysis of the variables was conducted, calculating absolute and relative frequencies for qualitative variables, and determining the arithmetic mean, standard deviation, minimum, and maximum values for quantitative variables. Additionally, a 95% confidence interval was estimated to provide a measure of uncertainty. Categorical variables were compared using the Chi-square test when all expected frequencies were ≥5. When any expected frequency was <5, Fisher’s exact test was used instead, as it provides more accurate results in cases with small sample sizes or sparse data. Twelve-month progression-free survival (PFS12) was defined as the duration from the initiation of therapy until either disease progression or death from any cause within 12 months. Patients who remained progression-free at their last follow-up or those who initiated a new treatment regimen (other than cetuximab) were censored in the analysis; for these patients, the censoring date was set at the start of the new treatment. In cases where there was no documented disease progression or death, participants were censored at their last known contact date, confirmed to be alive and progression-free. Survival probabilities were estimated using the Kaplan–Meier method, and the log-rank test was utilized to assess differences between subgroups, specifically evaluating the impact of KIR genotypes, haplotypes, and semi-haplotypes on survival outcomes. All statistical tests were two-tailed, with significance established at *p* < 0.05. To adjust for multiple comparisons and reduce the risk of type I error, the Bonferroni correction was applied where appropriate. Data collection, processing, and analysis were performed using SPSS version 25 statistical software (IBM, New York, NY, USA).

## Figures and Tables

**Figure 1 ijms-26-08062-f001:**
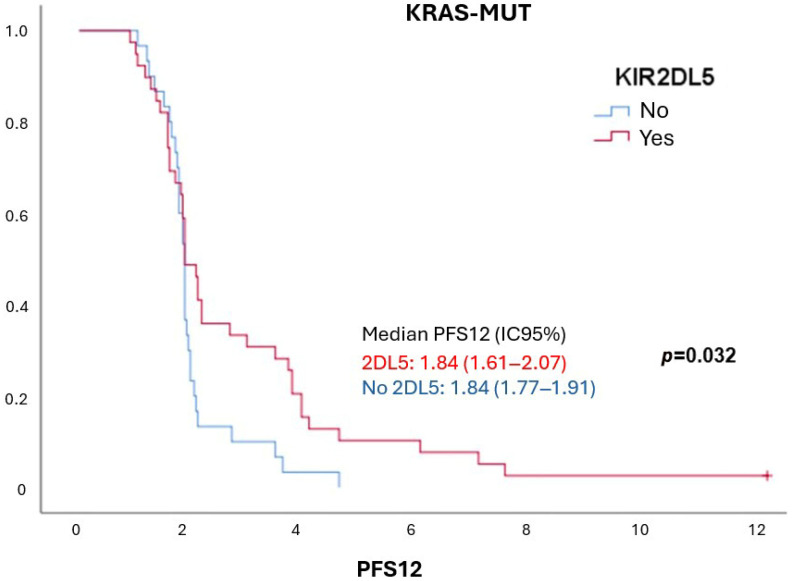
**Progression-free survival in KRAS-mutant colorectal cancer patients according to KIR2DL5 status.** The presence of KIR2DL5 was associated with longer twelve-month progression-free survival (PFS12) in KRAS-mutant patients treated with cetuximab.

**Figure 2 ijms-26-08062-f002:**
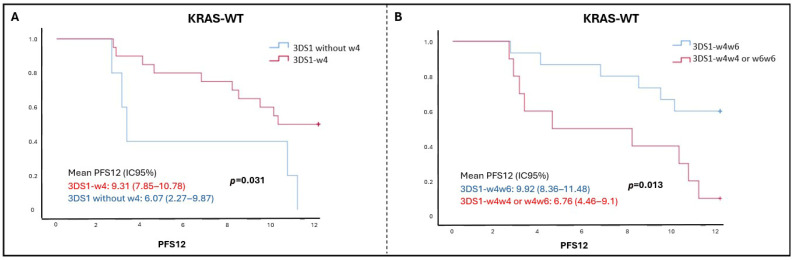
**Progression-free survival in KRAS wild-type colorectal cancer patients according to KIR–ligand combinations.** (**A**) The combination of KIR3DS1 with the Bw4 ligand was associated with longer twelve-month progression-free survival (PFS12) in KRAS wild-type (WT) patients treated with cetuximab. (**B**) Among KRAS-WT patients with KIR3DS1, those who were heterozygous for Bw4 (Bw4Bw6) showed longer PFS12 compared to Bw4 homozygotes (Bw4Bw4) or Bw6 homozygotes (Bw6Bw6).

**Figure 3 ijms-26-08062-f003:**
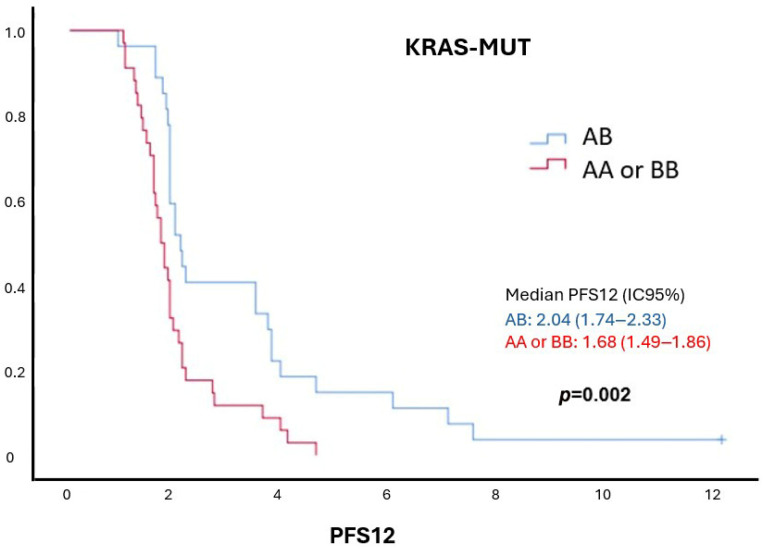
**Survival in KRAS-mutant colorectal cancer patients according to KIR haplotypes.** Among KRAS-mutant patients treated with cetuximab, those with heterozygous (AB) KIR haplotypes, comprising one activating (B) and one inhibitory (A) haplotype, demonstrated significantly longer twelve-month progression-free survival (PFS12) compared to homozygous patients (AA or BB).

**Table 1 ijms-26-08062-t001:** Baseline characteristics of patients included in the study.

Variable	KRAS-WT (n = 55) (%)	KRAS-Mutant (n = 69) (%)
Gender (n, %)	Male: 39 (70.9) Female: 16 (29.1)	Male: 35 (50.7) Female: 34 (49.3)
Age (median, range)	64 (41–84)	64 (42–82)
ECOG (n, %)	0: 24 (43.6) 1: 22 (40) 2: 3 (5.5) Unknown: 6 (10.9)	0: 12 (17.4) 1: 51 (73.9) 2: 6 (8.7)
Primary site (n, %)	Right colon: 16 (29.1) Left colon: 21 (38.2) Rectum: 18 (32.7)	Right colon: 16 (23.2) Left colon: 35 (50.7) Rectum: 18 (26.1)
Laterality (n, %)	Right-sided: 16 (29.1) Left-sided: 39 (70.9)	Right-sided: 16 (23.2) Left-sided: 53 (76.8)
Number of metastatic sites (n, %)	1: 28 (50.9) 2: 19 (34.5) 3 or more: 8 (14.5)	1: 26 (37.7) 2: 27 (39.1) 3 or more: 16 (23.2)
Previous treatments (n, %)	Yes: 30 (54.5) No: 24 (43.6) Unknown: 1 (1.8)	Yes: 69 (100) No: 0 (0)
Treatment time, months (median, range)	8.1 (1.9–12)	2 (0.5–12)
Number of treatment cycles (median, range)	17 (3–24)	4 (1–24)
Reason for end of treatment (n, %)	Progression: 28 (73.7) Death: 1 (2.6) Toxicity: 1 (2.6) Other: 8 (21.1)	Progression: 59 (86.8) Death: 2 (2.9) Toxicity: 4 (5.9) Other: 3 (4.4)
Progression at 12 months (n, %)	Yes: 33 (60) No: 22 (40)	Yes: 68 (92.8) No: 1 (1.4)
Exit at 12 months (n, %)	Yes: 14 (25.5) No: 41 (74.5)	Yes: 51 (73.9) No: 18 (26.1)
Exit reason (n, %)	Progression: 8 (57.2) Adverse event: 1 (7.1) Unknown: 5 (35.7)	Progression: 42 (82.4) Adverse event: 2 (3.9) Unknown: 7 (13.7)

**Table 2 ijms-26-08062-t002:** Frequency of KIR genes and their HLA ligands in KRAS-WT and KRAS-mutant patients.

KIR Genes and HLA Ligands	KRAS-WT (n = 55) (%)	KRAS-Mutant (n = 69) (%)	*p*-Value (X^2^/Fisher)	
KIR gene				
**2DL1** (n, %)	Positive: 54 (98.2) Negative: 1 (1.8)	Positive: 63 (91.3) Negative: 6 (8.7)	0.131	(Fisher)
**2DL2** (n, %)	Positive: 31 (56.4) Negative: 24 (43.6)	Positive: 37 (53.6) Negative: 32 (46.4)	0.761	(X^2^)
**2DL3** (n, %)	Positive: 53 (96.4) Negative: 2 (3.6)	Positive: 62 (89.9) Negative: 7 (10.1)	0.296	(Fisher)
**2DL5** (n, %)	Positive: 32 (58.2) Negative: 23 (41.8)	Positive: 39 (56.5) Negative: 30 (43.5)	0.853	(X^2^)
**3DL1** (n, %)	Positive: 50 (90.9) Negative: 5 (9.1)	Positive: 62 (89.9) Negative: 7 (10.1)	0.844	(X^2^)
**2DS1** (n, %)	Positive: 25 (45.5) Negative: 30 (54.5)	Positive: 37 (53.6) Negative: 32 (46.4)	0.366	(X^2^)
**2DS2** (n, %)	Positive: 31 (56.4) Negative: 24 (43.6)	Positive: 36 (52.2) Negative: 33 (47.8)	0.642	(X^2^)
**2DS3** (n, %)	Positive: 19 (34.5) Negative: 36 (65.5)	Positive: 29 (42) Negative: 39 (56.5) Unknown: 1 (1.4)	0.440	(X^2^)
**2DS4** (n, %)	Positive: 50 (90.9) Negative: 5 (9.1)	Positive: 63 (91.3) Negative: 6 (8.7)	1.000	(Fisher)
**2DS5** (n, %)	Positive: 19 (34.5) Negative: 36 (65.5)	Positive: 23 (33.3) Negative: 46 (66.7)	0.887	(X^2^)
**3DS1** (n, %)	Positive: 25 (45.5) Negative: 30 (54.5)	Positive: 27 (39.1) Negative: 42 (60.9)	0.478	(X^2^)
**2DP1** (n, %)	Positive: 54 (98.2) Negative: 1 (1.8)	Positive: 62 (89.9) Negative: 7 (10.1)	0.075	(Fisher)
**HLA ligand**				
**Bw4** (n, %)	Positive: 46 (83.6) Negative: 9 (16.4)	Positive: 46 (66.7) Negative: 21 (31.4)	0.056	(X^2^)
**C1** (n, %)	Positive: 45 (81.8) Negative: 10 (18.2)	Positive: 54 (80.6) Negative: 13 (19.4)	0.864	(X^2^)
**C2** (n, %)	Positive: 37 (67.3) Negative: 18 (32.7)	Positive: 48 (71.6) Negative: 19 (28.4)	0.601	(X^2^)

**Table 3 ijms-26-08062-t003:** Frequency of KIR–ligand combinations in KRAS-WT and KRAS-mutant patients.

KIR–Ligand Combinations	KRAS-WT n (%)	KRAS-Mutant n (%)	*p*-Value (X^2^)
**3DL1**–**Bw4**	Positive: 41 (82) Negative: 9 (18)	Positive: 41 (68.3) Negative: 19 (31.7)	0.101
**2DL1**–**C2**	Positive: 36 (66.7) Negative: 18 (33.3)	Positive: 43 (69.4) Negative: 19 (30.6)	0.757
**2DL2**–**C1**	Positive: 26 (83.9) Negative: 5 (16.1)	Positive: 27 (75) Negative: 9 (25)	0.373
**2DL3**–**C1**	Positive: 44 (83) Negative: 9 (17)	Positive: 50 (82) Negative: 11 (18)	0.883
**2DS1**–**C2**	Positive: 17 (68) Negative: 8 (32)	Positive: 26 (72.2) Negative: 10 (27.8)	0.722
**3DS1**–**Bw4**	Positive: 20 (80) Negative: 5 (20)	Positive: 16 (59.3) Negative: 11 (40.7)	0.105

**Table 4 ijms-26-08062-t004:** Frequency of genotypes and haplotypes in KRAS-WT and KRAS-mutant patients.

Genotypes and Haplotypes	KRAS-WT n (%)	KRAS-Mutant n (%)	*p*-Value (X^2^)
**AA**	Positive: 12 (21.8) Negative: 43 (78.2)	Positive: 16 (26.2) Negative: 45 (73.8)	0.579
**AB**	Positive: 30 (54.5) Negative: 25 (45.5)	Positive: 27 (44.3) Negative: 34 (55.7)	0.269
**BB**	Positive: 13 (23.6) Negative: 42 (76.4)	Positive: 18 (29.5) Negative: 43 (70.5)	0.475
**CENA/TELA**	Positive: 12 (21.8) Negative: 43 (78.2)	Positive: 16 (26.2) Negative: 45 (73.8)	0.579
**CENA/TELB**	Positive: 12 (21.8) Negative: 43 (78.2)	Positive: 12 (19.7) Negative: 49 (80.3)	0.776
**CENB/TELA**	Positive: 18 (32.7) Negative: 37 (67.3)	Positive: 14 (23) Negative: 47 (77)	0.239
**CENB/TELB**	Positive: 13 (23.6) Negative: 42 (76.4)	Positive: 19 (31.1) Negative: 42 (68.9)	0.366

**Table 5 ijms-26-08062-t005:** Median progression-free survival at 12 months (PFS12) according to the frequency of KIR genes and their HLA ligands in KRAS-WT and KRAS-mutant patients.

	KRAS-WT		KRAS-Mutant	
	PFS12 (IC 95%)	*p*-Value (Log Rank)	PFS12 (IC 95%)	*p*-Value (Log Rank)
**2DS1**	Positive (25): 10.15 (8.11–12.19) Negative (30): 8.84 (5.56–12.12)	0.955	Positive (37): 1.84 (1.81–1) Negative (32): 1.84 (1.7–1.98)	0.890
**2DS2**	Positive (31): 8.71 (7.33–10.1) Negative (24): 8.43 (7.02–0.83)	0.444	Positive (36): 1.84 (1.62–2.07) Negative (33): 1.84 (1.79–1.89)	0.163
**2DS3**	Positive (19): 9.44 (7.96–10.92) Negative (36): 8.14 (6.88–9.42)	0.314	Positive (29): 1.83 (1.78–1.89) Negative (39): 1.84 (1.81–1.87)	0.273
**2DS4**	Positive (50): 8.34 (7.28–9.40) Negative (5): 11.06 (10.03–12.08)	0.276	Positive (63): 1.84 (1.81–1.87) Negative (6): 1.84 (0.85–2.83)	0.800
**2DS5**	Positive (19): 10.15 (7.72–12.58) Negative (36): 8.84 (6.21–11.47)	0.751	Positive (23): 1.84 (1.81–1.87) Negative (46): 1.84 (1.75–1.93)	0.498
**3DS1**	Positive (25): 10.15 (8.11–12.19) Negative (30): 8.84 (5.56–12.12)	0.955	Positive (27): 1.84 (1.81–1.87) Negative (42): 1.84 (1.72–1.96)	0.968
**2DL1**	Positive (54): 10.15 (7.58–12.72) Negative (1): 3.02 (0–3.02)	0.031	Positive (63): 1.84 (1.81–1.87) Negative (6): 2.04 (0.78–3.29)	0.727
**2DL2**	Positive (31): 8.71 (7.33–10.1) Negative (24): 8.43 (7.03–9.83)	0.444	Positive (37): 1.84 (1.61–2.07) Negative (32): 1.84 (1.79–1.89)	0.232
**2DL3**	Positive (53): 10.15 (8.09–12.22) Negative (2): 3.02 (1.29–13.73)	0.944	Positive (62): 1.84 (1.81–1.87) Negative (7): 2.04 (0.86–3.22)	0.585
**2DL5**	Positive (32): 10.15 (8.42–11.88) Negative (23): 8.84 (5.01–12.67)	0.816	Positive (39): 1.84 (1.61–2.07) Negative (30): 1.84 (1.77–1.91)	** 0.032 **
**3DL1**	Positive (50): 8.34 (7.28–9.40) Negative (5): 11.06 (10.03–12.84)	0.276	Positive (62): 1.84 (1.81–1.87) Negative (7): 1.84 (1.75–1.92)	0.946
**2DP1**	Positive (54): 10.15 (7.58–12.72) Negative (1): 3.02 (0–3.02)	0.031	Positive (62): 1.84 (1.81–1.87) Negative (7): 2.04 (0.86–3.22)	0.976
**Bw4**	Positive (46): 8.85 (7.81–9.91) Negative (9): 7.23 (4.6–9.85)	0.064	Positive (46): 1.84 (1.78–1.89) Negative (21): 1.84 (1.6–2.04)	0.620
**C1**	Positive (45): 9.96 (7.62–12.29) Negative (10): 10.42 (7.34–13.49)	0.826	Positive (54): 1.84 (1.79–1.89) Negative (13): 1.84 (1.69–1.99)	0.198
**C2**	Positive (37): 10.58 (9.07–12.09) Negative (18): 4.79 (0–12.17)	0.219	Positive (48): 1.84 (1.75–1.94) Negative (19): 1.81 (1.75–1.86)	0.935

**Table 6 ijms-26-08062-t006:** Median PFS12 according to centromeric or telomeric haplotype in KRAS-WT and KRAS-mutant patients.

	KRAS-WT		KRAS-Mutant	
	PFS12 (IC 95%)	*p*-Value (Log Rank)	PFS12 (IC 95%)	*p*-Value (Log Rank)
**CENA/TELA** **Other**	(n = 12) 8.81 (8.64–8.97) (n = 43) 10.42 (8.22–12.61)	0.571	(n = 16) 1.74 (1.61–1.87) (n = 45) 1.84 (1.74–1.94)	**0.011**
**CENA/TELA + CENB/TELB** **CENA/TELB + CENB/TELA**	(n = 25) 10.15 (6.68–13.63) (n = 30) 9.96 (6.56–13.35)	0.919	(n = 35) 1.74 (1.51–1.97) (n = 26) 2.04 (1.74–2.33)	**0.002**

**Table 7 ijms-26-08062-t007:** Univariate and multivariate analyses of KRAS-WT patients (n = 55).

	Univariate Analysis			Multivariate Analysis		
Variable	Frequency	HR (95% CI)	*p*-Value	Frequency	HR (95% CI)	*p*-Value
Gender						
Female	16 (29.1%)	1	0.877	16 (29.1%)	1	0.682
Male	39 (70.9%)	0.94 (0.45–1.97)		39 (70.9%)	1.18 (0.54–2.58)	
**Median Age**						
<64 years	25 (45.5%)	1	0.788	25 (45.5%)	1	0.477
>64 years	30 (54.5%)	0.91 (0.46–1.81)		30 (54.5%)	0.12 (0.04–0.37)	
**ECOG**						
0	24 (43.6%)	1		-	-	-
1	22 (40%)	0.56 (0.17–1.19)	0.136	-	-	-
2	3 (5.5%)	0.46 (0.06–3.47)	0.453	-	-	-
**Number of cycles**						
<6	5 (9.1%)	**1**	**0**	5 (9.1%)	**1**	**0**
>6	50 (90.9%)	**0.16 (0.06–0.43)**		50 (90.9%)	**0.13 (0.04–0.39)**	
**Laterality**						
Right-sided	16 (29.1%)	1	0.849	-	-	-
Left-sided	39 (70.9%)	1.07 (0.51–2.26)		-	-	-
**Number of Metastatic sites**						
1	28 (50.9%)	1	0.844 0.837	-	-	-
2	19 (34.5%)	1.08 (0.51–2.28)		-	-	-
>3	8 (14.5%)	1.11 (0.41–3.04)		-	-	-
**KIR2DL1**						
No	1 (1.8%)	1	0.065	-	-	-
Yes	54 (98.2%)	0.14 (0.02–1.97)		-	-	-
**KIR2DL2**						
No	24 (43.6%)	1	0.446	-	-	-
Yes	31 (56.4%)	0.77 (0.38–1.52)		-	-	-
**KIR2DL3**						
No	2 (3.6%)	1	0.944	-	-	-
Yes	53 (96.4%)	1.07 (0.15–7.87)		-	-	-
**KIR2DL5**						
No	23 (41.8%)	1	0.816	-	-	-
Yes	32 (58.2%)	0.92 (0.46–1.84)		-	-	-
**KIR 2DS1**						
No	30 (54.5%)	1	0.955	-	-	-
Yes	25 (45.5%)	0.98 (0.49–1.95)		-	-	-
**KIR 2DS2**						
No	24 (43.6%)	1	0.446	-	-	-
Yes	31 (56.4%)	0.77 (0.39–1.52)		-	-	-
**KIR 2DS3**						
No	36 (65.5%)	1	0.317	-	-	-
Yes	19 (34.5%)	0.68 (0.33–1.44)		-	-	-
**KIR 2DS4**						
No	5 (9.1%)	1	0.289	-	-	-
Yes	50 (90.9%)	2.17 (0.52–9.08)		-	-	-
**KIR 2DS5**						
No	36 (65.5%)	1	0.751	-	-	-
Yes	19 (34.5%)	0.89 (0.43–1.84)		-	-	-
**KIR 3DL1**						
No	5 (9.1%)	1	0.289	-	-	-
Yes	50 (90.1%)	2.17 (0.52–9.08)		-	-	-
**KIR 3DS1**						
No	30 (54.5%)	1	0.955	-	-	-
Yes	25 (45.5%)	0.98 (0.49–1.95)		-	-	-
**KIR 2DP1**						
No	1 (1.8%)	1	0.065	-	-	-
Yes	54 (98.2%)	0.14 (0.017–1.13)		-	-	-
				-	-	-
**KIR3DS1–Bw4**						
No	5 (20%)	1	**0.026**	5 (20%)	1	**0.033**
Yes	20 (80%)	**0.29 (0.1–0.86)**		20 (80%)	**0.49 (0.25–0.94)**	
**3DS1–HLAB**						
Heterozygous (w4w6)	15 (27.3%)	1	**0.019**	-	1	**0.013**
Homozygous (w4w4 or w6w6)	10 (18.2%)	**3.48 (1.23–9.86)**		-	**2.22 (1.19–4.16)**	
**Genotype**						
AB	30 (54.5%)	1		-	-	-
AA	12 (21.8%)	1.9 (0.81–4.48)	0.143	-	-	-
BB	13 (23.6%)	0.92 (0.38–2.23)	0.849	-	-	-
**Genotype**						
AB	30 (54.5%)	1	0.517	-	-	-
AA or BB	25 (45.4%)	1.26 (0.62–2.56)		-	-	-
**Semi-haplotype**						
CENA/TELB or CENB/TELA	30 (54.5%)	1	0.108	-	-	-
CENA/TELA	12 (21.8%)	1.95 (0.86–4.41)		-	-	

**Table 8 ijms-26-08062-t008:** Univariate and multivariate analyses of KRAS-mutant patients (n = 69).

	Univariate Analysis			Multivariate Analysis		
Variable	Frequency	HR (95% CI)	*p*-Value	Frequency	HR (95% CI)	*p*-Value
Gender						
Female	34 (49.3%)	1	0.629	34 (49.3%)	1	0.71
Male	35 (50.7%)	0.89 (0.55–1.44)		35 (50.7%)	1.11 (0.65–1.88)	
**Median Age**						
<64 years	33 (47.8%)	1	0.178	33 (47.8%)	1	0.47
>64 years	36 (52.2%)	1.41 (0.85–2.34)		36 (52.2%)	1.01 (0.98–1.04)	
**ECOG**						
0	12 (17.4%)	1		-	-	-
1	51 (73.9%)	1.09 (0.58–2.05)	0.792	-	-	-
2	6 (8.7%)	1.26 (0.46–3.47)	0.660	-	-	-
**Number of cycles**						
<6	55 (79.7%)	**1**	**0**	-	-	-
>6	14 (20.3%)	**0.24 (0.12–0.45)**		-	-	
**Laterality**						
Right-sided	16 (23.2%)	1	0.71	-	-	-
Left-sided	53 (76.8%)	1.11 (0.63–1.97)		-	-	-
**Number of Metastatic sites**						
1	28 (50.9%)	1	0.38 0.09	28 (50.9%)	1	0.47 0.002
2	19 (34.5%)	1.28 (0.74–2.22)		19 (34.5%)	1.27 (0.67–2.40)	
>3	8 (14.5%)	1.75 (0.91–3.37)		8 (14.5%)	2.94 (1.46–5.92)	
**KIR2DL1**						
No	6 (8.7%)	1	0.736	-	-	-
Yes	63 (91.3%)	1.16 (0.49–2.69)		-	-	-
**KIR2DL2**						
No	32 (46.4%)	1	0.247	-	-	-
Yes	37 (53.6%)	0.75 (0.46–1.22)		-	-	-
**KIR2DL3**						
No	7 (10.1%)	1	0.598	-	-	-
Yes	62 (89.9%)	1.24 (0.56–2.74)		-	-	-
**KIR2DL5**						
No	30 (43.5%)	1	**0.039**	-	-	-
Yes	39 (56.5%)	**0.59 (0.35–0.97)**		-	-	-
**KIR2DS1**						
No	32 (46.4%)	1	0.893	-	-	-
Yes	37 (53.6%)	0.97 (0.6–1.56)		-	-	-
**KIR2DS2**						
No	33 (47.8%)	1	0.178	-	-	-
Yes	36 (52.2%)	0.71 (0.44–1.17)		-	-	-
**KIR2DS3**						
No	39 (56.5%)	1	0.136	-	-	-
Yes	29 (42%)	0.69 (0.43–1.12)		-	-	-
**KIR2DS4**						
No	6 (8.7%)	1	0.806	-	-	-
Yes	63 (91.3%)	1.11 (0.48–2.59)		-	-	-
**KIR2DS5**						
No	46 (66.7%)	1	0.512	-	-	-
Yes	23 (33.3%)	0.84 (0.51–1.4)		-	-	-
**KIR3DL1**						
No	7 (10.1%)	1	0.947	-	-	-
Yes	62 (89.9%)	0.97 (0.44–2.14)		-	-	-
**KIR3DS1**						
No	42 (60.9%)	1	0.969	-	-	-
Yes	27 (39.1%)	1.01 (0.62–1.64)		-	-	-
**KIR2DP1**						
No	7 (10.1%)	1	0.977	-	-	-
Yes	62 (89.9%)	1.01 (0.46–2.22)		-	-	-
				-	-	-
**KIR3DS1–Bw4**						
No	11 (40.7%)	1	0.618	-	-	-
Yes	16 (59.3%)	0.82 (0.38–1.79)		-	-	-
**3DS1–HLAB**						
Heterozygous (w4w6)	20 (28.9%)	1	0.598	-	-	-
Homozygous (w4w4 or w6w6)	7 (10.1%)	1.28 (0.51–3.22)		-	-	-
**Genotype**						
AB	27 (44.3%)	1		27 (44.3%)	1	
AA	16 (26.2%)	**2.27 (1.21–4.25)**	**0.01**	16 (26.2%)	**2.58 (1.31–5.10)**	**0.006**
BB	18 (29.5%)	**1.99 (1.12–3.55)**	**0.02**	18 (29.5%)	**1.93 (1.04–3.58)**	**0.03**
**Genotype**						
AB	27 (44.3%)	1	**0.004**	-	**1**	**0.005**
AA or BB	34 (55.7%)	**2.10 (1.28–3.47)**		-	**2.16 (1.26–3.78)**	
**Semi-haplotype**						
CENA/TELB or CENB/TELA	26 (37.7%)	1	**0.03**	-	**1**	**0.04**
CENA/TELA	16 (26.2%)	**2.31 (1.08–4.95)**		-	**2.50 (1.02–6.14)**	

## Data Availability

The raw data supporting the conclusions of this article will be made available by the authors upon reasonable request, without undue reservation, and in compliance with applicable ethical and legal guidelines.

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
