# Peer review of "Immune Modulation Through KIR–HLA Interactions Influences Cetuximab Efficacy in Colorectal Cancer"

_ijms, 2025, doi:10.3390/ijms26168062_

Round 1

Reviewer 1 Report

Comments and Suggestions for Authors

The work entitled "Immune Modulation Through KIR/HLA Interactions Influences Cetuximab Efficacy in Colorectal Cancer" is an attempt to compare the effect of KIR/HLA interactions on the efficacy of cetuximab in patients with colorectal cancer. The authors analyze two subgroups: one with KRAS mutations and the other with wild-type KRAS. The work itself presents a well-developed genotypic and haplotype approach, thanks to which a broad spectrum of analyses provides greater research credibility. The text of the manuscript itself has its own coherent and logical structure. However, despite the above advantages, I have several comments and suggestions for the authors, which I would appreciate their explanation. I present them below:

1. You analyzed many variables, if I counted correctly 15 KIR genes, ligand interactions and haplotypes, in this situation there is a risk of obtaining false positive results, my question is whether you used Bonferroni correction in multiple analyses.

2. Another issue is the attempt to calculate the statistical power of the tests and verify whether the number of patients studied is sufficient to confirm the conclusions presented in the work.

3. I would also like to ask the authors whether it would not be worth performing analyses that would take into account clinical variables such as tumor location (e.g., right-sided vs. left-sided), MSI status, other mutations (NRAS, BRAF), line of treatment, which could affect the response to cetuximab and/or immunological interactions.

4. Do the authors have the possibility to perform functional tests of NK cells and confirm the biological significance of the observed correlations?

To sum up, the work presents a valuable analysis. After obtaining the answer to the above question, I will most definitely recommend publishing this interesting work.

Author Response

Comments 1. You analyzed many variables, if I counted correctly 15 KIR genes, ligand interactions and haplotypes, in this situation there is a risk of obtaining false positive results, my question is whether you used Bonferroni correction in multiple analyses.

Response 1.- Thank you for your comment. We applied the Bonferroni correction where appropriate to account for multiple comparisons and reduce the risk of false positives. This has been clarified in the revised Methods section.

Comments 2. Another issue is the attempt to calculate the statistical power of the tests and verify whether the number of patients studied is sufficient to confirm the conclusions presented in the work.

Response 2.- Thank you for your question. As this is a retrospective study, the number of patients included was fixed and based on available data, so no formal statistical power calculation was performed beforehand. Despite this, our analysis provides valuable insights within the scope of the study, and we believe that our results contribute meaningful information to the field. Future studies with larger cohorts could help further validate these findings.

Comments 3. I would also like to ask the authors whether it would not be worth performing analyses that would take into account clinical variables such as tumor location (e.g., right-sided vs. left-sided), MSI status, other mutations (NRAS, BRAF), line of treatment, which could affect the response to cetuximab and/or immunological interactions.

Response 3.- We fully agree with the reviewer that these clinical parameters can influence both the response to cetuximab and immunological interactions. However, the current study was designed with a primary focus on exploring the immunogenetic influence of KIR/HLA interactions on cetuximab efficacy, particularly within the framework of KRAS mutational status. While clinical data such as tumor location, MSI status, and additional mutational information would undoubtedly enrich the analysis, these variables were not uniformly available across our study cohort, which precluded their inclusion without introducing potential selection bias or reducing the statistical power due to incomplete data. In addition, our intention was to maintain a genetically driven immunological focus to provide a proof-of-concept regarding the modulatory role of KIR/HLA combinations. We consider this a foundational step that could be followed by future studies incorporating more granular clinical data and multivariable models. We are currently working on expanding our dataset to address these factors in subsequent analyses.

Comments 4.  Do the authors have the possibility to perform functional tests of NK cells and confirm the biological significance of the observed correlations?

Response 4.- We recognize the importance of complementing genotypic correlations with functional validation. However, this study was conducted retrospectively using biobanked DNA samples obtained from peripheral blood, and no viable PBMCs were available for functional NK cell assays. The absence of viable biological material limits our capacity to directly assess NK cell functionality in this specific cohort. Nevertheless, the associations we observed between KIR/HLA genotypes and clinical outcomes are supported by existing mechanistic insights from the literature on NK cell biology. We believe that our findings provide a compelling rationale for future prospective studies that will incorporate both functional NK cell profiling and detailed clinical annotation.

We hope the reviewer finds these clarifications satisfactory and appreciates the rationale behind our methodological choices in the context of the current study design.

Reviewer 2 Report

Comments and Suggestions for Authors

This manuscript presents a well-structured and comprehensive study on the role of KIR/HLA interactions in modulating cetuximab efficacy in metastatic colorectal cancer, with stratification by KRAS status. The topic is timely and relevant, combining immunogenetics with therapeutic outcomes.

Suggestions:

1. The manuscript is lengthy and could benefit from condensing the Results section.
2. The Discussion occasionally repeats earlier points and would be strengthened by a more focused comparison with previous studies.
3. Some figures (e.g., survival curves) could be presented more clearly with consistent labeling.
4. Add more details on how KIR/HLA typing was validated (e.g., internal controls, genotyping error rate).

Recommendation: Minor Revision

The manuscript is suitable for publication after minor revisions addressing clarity and conciseness.

Author Response

Comments 1. The manuscript is lengthy and could benefit from condensing the Results section.

Response 1. Thank you for your advice. In response, we have condensed the Results section, reducing its length by one full page while maintaining clarity in the presentation of the findings. These changes have been highlighted in the revised manuscript.

Comments 2. The Discussion occasionally repeats earlier points and would be strengthened by a more focused comparison with previous studies.

Response 2.- Thank you for your suggestion. We have now revised and refined the Discussion section to avoid repetition and improve focus. The updated version now provides a clearer, more concise comparison with previous studies, highlighting the novel aspects of our work. Modifications have been highlighted in the updated version of the manuscript.

Comments 3. Some figures (e.g., survival curves) could be presented more clearly with consistent labeling.

Response 3. Thank you for your observation. All figures have been revised to ensure consistent labeling and improved clarity. They now also incorporate key data from the survival analyses, providing a clearer and more informative presentation of the results.

Comments 4. Add more details on how KIR/HLA typing was validated (e.g., internal controls, genotyping error rate).

Response 4.- Thanks for your comment. KIR/HLA typing in our laboratory is performed under the framework of European Federation for Immunogenetics (EFI) accreditation, which ensures compliance with internationally recognised standards for Histocompatibility & Immunogenetics testing. Each batch of determinations includes internal positive and negative controls to verify assay performance and specificity. In addition, our laboratory participates in the GECLID external quality assessment programme for both HLA typing and KIR genotyping. This programme, sponsored by the Spanish Society of Immunology and the Iberian Society of Cytometry, is recognised as a prestigious External Quality Assurance scheme for diagnostic immunology laboratories. Over the past several years, we have consistently achieved a genotyping error rate of 0% in both HLA typing and KIR genotyping in these external quality assessments, further confirming the accuracy and reliability of our results. The corresponding details have now been incorporated into the Materials and Methods section to clarify this issue as requested by the referee. Changes have been highlighted in the revised manuscript.